# A Mediterranean Eating Pattern Combining Energy and Time-Restricted Eating Improves Vaspin and Omentin Concentrations Compared to Intermittent Fasting in Overweight Individuals

**DOI:** 10.3390/nu15245058

**Published:** 2023-12-09

**Authors:** Spyridon N. Karras, Theocharis Koufakis, Djordje S. Popovic, Lilian Adamidou, Paraskevi Karalazou, Katerina Thisiadou, Pantelis Zebekakis, Kali Makedou, Kalliopi Kotsa

**Affiliations:** 1Division of Endocrinology and Metabolism, First Department of Internal Medicine, Medical School, AHEPA University Hospital, Aristotle University of Thessaloniki, 1 St. Kiriakidi Street, 54636 Thessaloniki, Greece; karraspiros@yahoo.gr (S.N.K.); pzempeka@auth.gr (P.Z.); 2Second Propaedeutic Department of Internal Medicine, Aristotle University of Thessaloniki, Hippokration General Hospital, 54642 Thessaloniki, Greece; thkoyfak@auth.gr; 3Clinic for Endocrinology, Diabetes and Metabolic Disorders, Clinical Center of Vojvodina, 21000 Novi Sad, Serbia; pitstop021@gmail.com; 4Medical Faculty, University of Novi Sad, 21000 Novi Sad, Serbia; 5Department of Dietetics and Nutrition, AHEPA University Hospital, 54636 Thessaloniki, Greece; loula1924@hotmail.com; 6Laboratory of Biochemistry, AHEPA General Hospital, School of Medicine, Aristotle University of Thessaloniki, 54636 Thessaloniki, Greece; vivikarala@gmail.com (P.K.); thisiadou@yahoo.gr (K.T.); kalimakedou@gmail.com (K.M.)

**Keywords:** orthodox fasting, time-restricted eating, vaspin, omentin, nesfatin, visfatin

## Abstract

Athonian Orthodox fasting (AOF) is characterized by energy- and time-restricted eating (TRE) and is based on the Mediterranean diet. We aimed to investigate the impact of AOF compared to another TRE model on vaspin, omentin, nesfatin, and visfatin levels. We included 25 individuals (mean age 50.3 ± 8.6 years, 24% men) who practiced AOF and abstained from animal products, with the exception of seafood and fish. This group adopted a 12 h eating interval (08.00 to 20.00). In total, 12 participants (mean age 47.7 ± 8.7 years, 33.3% men) who practiced 16:8 TRE (eating from 09:00 to 17:00) and were allowed to consume meat served as the controls. Anthropometric and dietary data and adipokine levels were prospectively collected at three time points: at baseline, after the end of the diets (7 weeks), and 5 weeks after the participants returned to their typical eating habits (12 weeks from baseline). Vaspin levels decreased [795.8 (422.1–1299.4) (baseline) vs. 402.7 (203.8–818.9) (7 weeks) pg/mL, *p* = 0.002] and omentin levels increased [568.5 (437.7–1196.5) (baseline) vs. 659.0 (555.7–1810.8) (12 weeks) pg/mL, *p* = 0.001] in the AOF group, while none of the analyzed adipokines changed significantly in the TRE group. The variations observed in vaspin and omentin concentrations in the AOF group were independent of age, sex, changes in anthropometry and fat intake. In conclusion, AOF can significantly reduce vaspin and increase omentin, whose levels are known to increase and decrease, respectively, in obesity and type 2 diabetes. The implications of these findings for cardiometabolic health warrant further investigation.

## 1. Introduction

Christian Orthodox fasting (OF) is a type of religious fasting, followed by a large number of faithful around the world for approximately 180–200 days per year, aiming at purgation of the soul and the body [1]. During the OF periods, the consumption of animal products is restricted, while fish and seafood are occasionally allowed. The core components of this nutritional pattern are olive oil, grains, legumes, and fruits, thus sharing several common characteristics with the typical Mediterranean diet (MedDiet). Previous studies have shown that OF exerts positive effects on plasma lipids [2,3,4], body composition [5,6], and circulating levels of adipokines, including irisin [7] and adiponectin [8].

Athonian Orthodox fasting (AOF) is a subset of classic OF, followed by male monks who reside in the holy community of Mount Athos in northern Greece. AOF differentiates from typical OF with respect to meat consumption, which is totally prohibited in the former, even during non-fasting days. The protein intake in AOF is largely based on the consumption of fish and seafood, similar to the MedDiet, which typically includes 1–5 servings of fish per week [9]. Furthermore, AOF has a macronutrient composition similar to the MedDiet, specifically 50–60% carbohydrates, 15–20% protein, and 30% fat [10]. As we have previously demonstrated, AOF is characterized by a high intake of monounsaturated fat (MUFA) and polyunsaturated fatty acids [6,7,8], as happens with the MedDiet. Moreover, the AOF incorporates features of intermittent fasting, since monks abstain from food from sunset to dawn, due to their religious duties. Temperance is deeply integrated into the mentality of monastic life in Athos. This is reflected in the fact that compared to lay Orthodox fasters, monks have been shown to consume a lower amount of energy daily [11], resulting in an optimal glucose, lipid and anthropometric [i.e., body mass index (BMI) and body fat (BF) within the normal range] profile, and low insulin resistance [12].

Following dietary interventions that induce weight loss, such as fasting, circulating adipokine levels can change as a result of respective changes in adipose tissue. Vaspin, omentin, nesfatin, and visfatin are adipokines discovered nearly 20 years ago whose levels have been shown to correlate with the risk of developing obesity and type 2 diabetes (T2D) [13,14]. Vaspin has insulin-sensitizing effects, improving insulin tolerance in animal models of diabetes [15]. In humans, vaspin levels increase in the diabetic state and have been shown to present a positive correlation with the risk of macrovascular [16] and microvascular complications [17]. Omentin is secreted by visceral fat adipose tissue, enhancing insulin action in human adipocytes [18] and has been suggested to represent a novel link between inflammation, metabolic disturbance, and cardiovascular risk [19]. A meta-analysis indicated decreased omentin concentrations in subjects with T2D and gestational diabetes compared to normoglycemic controls [20], while different studies have reported an increase in its levels in obesity, possibly as a counterregulatory mechanism to protect against insulin resistance [21].

Nesfatin was discovered as a potent anorexigenic peptide in the rat hypothalamus, decreasing the motivational and rewarding value of food [22]. A systematic review suggested increased circulating nesfatin concentrations in people with a recent T2D diagnosis; in contrast, they were found to be lower than healthy controls in patients with T2D receiving glucose-lower treatment [23]. Visfatin exerts endocrine, autocrine, and paracrine actions that result in enhanced cell proliferation, synthesis of nicotinamide nucleotides, inhibition of hepatic glucose release, and stimulation of glucose utilization by peripheral tissues [24]. Its levels correlate with markers of systemic inflammation, beta cell function, intra-abdominal obesity, and atherosclerosis [25].

Considering the close association between adipokine levels and metabolic health, the effects of various dietary models—especially those that are gaining increasing popularity—on these molecules deserve to be investigated. Time-restricted eating (TRE) is based on the important role of circadian rhythms in physiology and disease, emerging as an alternative to calorie restriction to improve body weight, glucose profile, and cardiovascular risk factors [26]. In TRE, subjects abstain from food for 16 or 20 h during the day, while subtypes include early TRE (eating early during the day) and late TRE (eating late during the day) [27]. Although it has been a field of intense medical research in the last decade, a limited number of studies have investigated its impact on adipokine levels in humans, mostly showing no significant effects [28].

The research hypothesis to be tested in this study is that the composition of MedDiet is the main driver of its metabolic benefits. A dietary model that has consistently proven positive metabolic effects, such as TRE, would be the ideal control diet to test our hypothesis. However, the comparison between the two nutritional patterns would not be reasonable if the MedDiet did not include features of intermittent fasting. For this reason, we chose to compare AOF with TRE, since both diets include eating windows, although with slightly different time intervals (12 versus 8 h). Therefore, we prospectively assessed the impact of AOF on vaspin, omentin, nesfatin, and visfatin levels, compared to a TRE model, in overweight but otherwise metabolically healthy individuals (i.e., without a diagnosis of a chronic metabolic disease requiring medical surveillance).

## 2. Materials and Methods

### 2.1. Study Population

Adults aged 18–65 years who had a BMI between 25 and 29.9 kg/m^2^ were included in the study. According to the systematic review and meta-analysis by Silva et al. [29], better health is one of the main motivations for people to improve their weight. Therefore, we hypothesized that overweight individuals would be more motivated than normal weight peers to improve their health through diet [30]. Exclusion criteria were: (i) the presence of chronic kidney disease, severe liver disease, diagnosis of prediabetes (fasting glucose 100–125 mg/dL or glycated hemoglobin 5.7–6.4% or blood glucose 140–199 mg/dL at 2 h post 75 g glucose load) or diabetes mellitus (fasting glucose ≥ 126 mg/dL or glycated hemoglobin ≥ 6.5% or blood glucose ≥ 200 mg/dL at 2 h post 75 g glucose load), dyslipidemia, arterial hypertension, or uncontrolled hypothyroidism (not adequately controlled or first diagnosed and not treated), (ii) recent surgery or severe infections (during the past 3 months), (iii) administration of drugs that can alter body weight, glucose and lipid metabolism (e.g., statins, corticosteroids, antipsychotics), (iv) intake of vitamins or mineral supplements, (v) physical disabilities and/or neurodegenerative disorders that could affect physical activity, (vi) acute infections and chronic degenerative diseases.

### 2.2. Diets

The first group consisted of lay people who were asked to practice the Athonian type of fasting, while the TRE group followed a diet based on the recommendations of the American Heart Association (AHA) for overweight and obesity [31]. The AOF group adopted a 12 h eating interval (08.00 to 20.00), while the TRE group consumed food from 09:00 to 17:00. Outside the above time periods, only drinking water, tea or coffee was allowed. All participants followed a hypocaloric diet, providing a total of 5020–6276 kJ (1200–1500 kcal) per day for women and 6276–7531 kJ (1500–1800 kcal) per day for men. Their daily energy requirements were calculated based on their daily basal metabolic rate (assessed with the SC-330 S total body composition analyzer, Tanita Corporation, Tokyo, Japan using bioelectric impedance analysis [BIA]), adjusted for an expected weight loss of 0.5 kg per week (energy deficit of 2092–3138 kJ [500–750 kcal] per week) and estimated according to the AHA recommendations. The diet period lasted seven weeks (48 days) and took place during the fasting of Lent. Adherence to dietary plans was evaluated with a 3 day food record (two weekdays and one weekend day) at the end of the study period, while the Nutrition Analysis Software Food Processor [https://esha.com/products/food-processor/ (accessed on 2 August 2022)] was used to analyze food records. After the end of the diets, participants were advised to return to their typical dietary patterns until the 12 week re-evaluation.

The AOF group was not allowed to consume animal products (meat, poultry, eggs, dairy and cheese), with the exception of seafood and fish, which fasters were permitted to eat on two specific weekdays. The macronutrients in this group were estimated to be 45–55% carbohydrates [of total daily energy intake (TEI)], 10–20% protein and 30–40% fat. The control group was allowed to eat low-fat meat products with a dietary macronutrient distribution of 52–55% (of TEI) carbohydrates, 15–18% protein, and 30% fat. The recommended amounts in Orthodox fasters and TRE controls were estimated according to the principles of the Greek Orthodox Church fasting practice [4,5,6,7,8] and the Greek National Dietary Guidelines for Adults [32], respectively. Finally, all participants were asked to keep a stable level of physical activity during the study period, defined as 150 min per week of moderate intensity aerobic exercise, according to the AHA recommendations [31].

### 2.3. Anthropometric Measurements

Height was measured to the nearest 0.1 cm with a Holtain wall stadiometer. Body weight was recorded to the nearest 0.01 kg using a calibrated computerized digital balance (K-Tron P1-SR, Onrion LLC, Bergenfield, NJ, USA); each participant was barefoot and lightly dressed during the measurement. The BMI was calculated as the ratio of weight in kilograms divided by height in meters squared (kg/m^2^). Waist circumference (WC) was measured midway between the lowest rib and the iliac crest using an anthropometric tape. BF mass and percentage and lean body mass (LBM) were measured using BIA (SC-330 S, Tanita Corporation, Tokyo, Japan). All measurements were independently confirmed by a second investigator.

### 2.4. Biochemical Analysis

Blood samples were drawn in the morning, after a 12 h overnight fast by antecubital venipuncture, and the samples were stored at −20 °C prior to analysis. Visfatin was determined in serum using a sandwich ELISA kit (R&D Systems, Minneapolis, MN, USA) and the use of mouse anti-human PBEF capture antibody and biotinylated mouse anti-human PBEF detection antibody. Nesfatin was also determined in serum using a sandwich ELISA kit (R&D Systems, Minneapolis, MN, USA) and the use of sheep anti-human Nesfatin-1 capture antibody and biotinylated sheep anti-human Nesfatin-1 detection antibody. Omentin was assayed in serum with a precoated ELISA kit (BOSTER Picokine ELISA, Pleasanton, CA, USA) using a mouse anti-human monoclonal capture antibody and a biotinylated goat anti-human polyclonal antibody. Vaspin was determined in serum with a sandwich ELISA kit (ALPCO Diagnostics, Salem, NH, USA) using a rabbit anti-human polyclonal capture antibody and a biotinylated rabbit anti-human polyclonal antibody.

### 2.5. Statistical Analysis

Depending on their distribution, which was tested with the D’Agostino–Pearson test, the continuous variables are presented as mean ± standard deviation (SD) (normal distribution), and as median (interquartile range) (IR) (without a normal distribution). Frequencies of dichotomous variables (gender) are displayed as percentages. To test the differences in the baseline characteristics between AOF and TRE groups, we used the Student *t*-test (continuous variables with normal distribution), the Mann–Whitney test (continuous variables without a normal distribution), and the test of proportions (dichotomous variable). For the purpose of the analysis of changes in variables at the different study time points, we used the repeated measures analysis of variance (ANOVA). Since ANOVA represents the parametric statistical test, inclusion of variables that are not normally distributed may produce misleading results. Therefore, variables without a normal distribution underwent the logarithmic transformation prior to the analysis, with the assumption that transforming the data will make them fit the assumptions better. In order to assess whether variations in adipokine levels during the study period were significantly correlated with simultaneous changes in other parameters analyzed, we used multiple regression analysis. The threshold level for statistical significance was set at *p* < 0.05.

### 2.6. Ethical Aspects

All procedures in the present study were in accordance with the Declaration of Helsinki of 1964 and its subsequent amendments. The study protocol was approved by the institutional review board of the Aristotle University of Thessaloniki (approval number 25224/2019, approval date 14 August 2019), and the participants provided their written informed consent before enrolment. The personal information of the study participants was processed anonymously to ensure confidentiality.

## 3. Results

### 3.1. Demographic, Anthropometric and Dietary Data

In total, 25 individuals (mean age 50.3 ± 8.6 years, 24% men) were included in the AOF group and 12 individuals (mean age 47.7 ± 8.7 years, 33.3% men) were recruited into the TRE group. The two groups presented comparable baseline characteristics in terms of age (*p* = 0.39), gender distribution (*p* = 0.84), BMI (AOF: 29.3 ± 5.6 vs. TRE: 29.1 ± 6.9 kg/m^2^, *p* = 0.93), BF (AOF: 36.3 ± 8.5 vs. TRE: 32.8 ± 7.8%, *p* = 0.24), LBM (AOF: 47.8 ± 9.9 vs. TRE: 53.2 ± 12.7 kg, *p* = 0.16), and WC (AOF: 93.4 ± 14.0 vs. TRE: 94.9 ± 16.5 cm, *p* = 0.78).

Participants in both groups experienced a significant reduction in BMI at 7 and 12 weeks compared to the baseline as a result of the diets: AOF: 29.3 ± 5.6, 28.5 ± 5.3, 28.3 ± 5.3 kg/m^2^, *p* < 0.001 (95% confidence interval (CI): 0.46–1.19) and *p* = 0.004 (95% CI: 0.29–1.73), respectively; TRE: 29.1 ± 6.9, 28.3 ± 6.5, 28.0 ± 6.7 kg/m^2^, *p* = 0.02 (95% CI: 0.11–1.47) and *p* < 0.001 (95% CI: 0.52–1.66), respectively. Additionally, a significant decrease in BF values at 12 weeks compared to the baseline was evident in both groups: AOF: 32.9 ± 6.9 vs. 36.3 ± 8.5%, *p* = 0.01 (95% CI: 0.63–6.07) and TRE: 30.5 ± 8.0 vs. 32.8 ± 7.8%, *p* = 0.03 (95% CI: 0.25–4.39). WC was significantly reduced during the study only in the TRE group: 94.9 ± 16.5 (baseline) vs. 91.5 ± 16.1 (7 weeks) vs. 90.5 ± 15.8 (12 weeks) cm, *p* = 0.01 (95% CI: 0.83–6.00) and *p* = 0.007 (95% CI: 1.25–7.59), respectively.

In the AOF group, the daily intake of total fat (TF) and saturated fat (SFA) decreased between the baseline and 7 weeks (*p* < 0.001 and *p* < 0.0001, respectively) and then increased between 7 and 12 weeks (*p* = 0.01 and *p* < 0.0001, respectively). The same pattern was also evident in changes in TF and SFA intake for the TRE group between the baseline and 7 weeks (*p* < 0.001 for both comparisons), and between 7 and 12 weeks (*p* = 0.02 and *p* = 0.005, respectively). The intake of monounsaturated fat (MUFA) increased in the AOF group between the baseline and 7 weeks (*p* < 0.0001) and then decreased between 7 and 12 weeks (*p* < 0.0001). We did not observe significant changes in SFA intake in the TRE group and fiber intake in either group during the study. Table 1 presents the changes in anthropometric, biochemical and dietary parameters in the AOF and TRE groups throughout the study.

### 3.2. Adipokine Concentrations

In the AOF group, we observed a significant decrease in vaspin levels at 7 weeks compared to baseline [402.7 (203.8–818.9) vs. 795.8 (422.1–1299.4) pg/mL, *p* = 0.002]. After cessation of fasting, vaspin concentrations increased significantly from 7 to 12 weeks [402.7 (203.8–818.9) vs. 612.5 (474.7–1134.9) pg/mL, *p* = 0.004]. Omentin concentrations increased significantly at 12 weeks compared to the 7 week and baseline values [659.0 (555.7–1810.8) vs. 489.4 (397.5–1245.4) and 568.5 (437.7–1196.5) pg/mL, respectively, *p* = 0.002 and *p* = 0.001, respectively].

It should be noted that changes in vaspin concentrations were independent of age (*p* = 0.80), sex (*p* = 0.72), variations in BMI (*p* = 0.42), and intake of TF (*p* = 0.41), SFA (*p* = 0.45), and MUFA (*p* = 0.73). Similarly, changes in omentin values were independent of age (*p* = 0.24), sex (*p* = 0.80), changes in BF (*p* = 0.92) and BMI (*p* = 0.35), and intake of TF (*p* = 0.88), SFA (*p* = 0.83) and MUFA (*p* = 0.61). No significant changes in nesfatin [380.9 ± 654.1 vs. 387.9 ± 667.2 vs. 395.1 ± 680.1 ng/mL, *p* = 1.00 and *p* = 1.00, respectively] and visfatin [3.6 (2.8–16.7) vs. 3.7 (3.1–10.8) vs. 3.2 (2.9–11.2) ng/mL, *p* = 1.00 and *p* = 0.39, respectively] levels were documented during the study.

In the TRE group, the concentrations of omentin [622.1 (441.4–1059.2) vs. 567.5 (446.5–806.2) vs. 698.2 (564.8–1267.5) pg/mL, *p* = 0.77 and *p* = 0.94, respectively], vaspin [641.6 (322.8–1519.3) vs. 715.4 (257.8–1556.9) vs. 855.3 (527.0–1488.3) pg/mL, *p* = 1.00 and *p* = 0.90, respectively), nesfatin [419.5 ± 690.3 vs. 452.1 ± 699.9 vs. 464.0 ± 622.0 ng/mL, *p* = 0.95 and *p* = 1.00, respectively] and visfatin [3.4 (3.1–7.5) vs. 3.2 (2.9–4.7) vs. 3.5 (3.0–8.9) ng/mL, *p* = 0.20 and *p* = 1.00, respectively) remained unchanged during the study period (Figure 1).

## 4. Discussion

To the best of our knowledge, this is the first study to assess the effects of AOF on circulating concentrations of vaspin, omentin, nesfatin, and visfatin. We demonstrated significant reductions in vaspin and increases in omentin, whose levels are known to increase and decrease, respectively, in obesity and T2D. On the contrary, we did not observe significant variations in nesfatin and visfatin, while none of the examined adipokines changed significantly during the study in a group of participants who practiced TRE.

We have previously demonstrated that OF, a dietary model that shares several characteristics with the MedDiet, can increase the circulating levels of irisin [7] and adiponectin [8]. Mansour et al. have shown that the adoption of the MedDiet for 6 months decreases vaspin concentrations among women with obesity and T2D [33]. Chang et al. demonstrated that losing at least 2% of baseline weight with a combination of lifestyle change and administration of the antiobesity agent orlistat resulted in a decrease in vaspin levels [34]. Handisurya et al. showed similar improvements in vaspin concentrations in people with morbid obesity (mean BMI 47.12 kg/m^2^) who underwent Roux-en-Y gastric bypass surgery [35]. In accordance with our findings, previous work showed that weight loss secondary to a hypocaloric nutritional model based on the MedDiet leads to increased omentin levels [36]. A different study showed that after laparoscopic sleeve gastrectomy, omentin concentrations increase and omentin mRNA expression in subcutaneous fat is down-regulated among subjects with obesity [37]. However, in all of the aforementioned studies, the improvements in omentin and vaspin were a consequence of hypocaloric diets, anti-obesity medications, or surgical interventions (or a combination of the above) that induced weight loss. In our study, improvements in adipokine levels were found to be independent of changes in anthropometry.

Another parameter that could potentially affect the circulating concentrations of vaspin and omentin is chrononutrition. The term “chrononutrition” refers to a relatively new field of health sciences that aims to understand how the time of eating affects human health [38]. Serum vaspin levels increase before meals and decrease within 2 h after eating, present a nocturnal rise when its levels are greater by approximately 250% compared to mid-afternoon nadir concentrations, and are positively affected by a 20 h fast [39]. Ebrahimi et al. have demonstrated that Ramadan fasting, a religious fasting model in which fasters abstain from eating from dawn to sunset, induces significant changes in vaspin and omentin among subjects with nonalcoholic fatty liver disease [40]. However, in the present study, in which participants in both groups adopted TRE (although with different eating windows), vaspin and omentin improved significantly only in the AOF group. Sutton et al. demonstrated that early TRE (a 6 h eating window with dinner before 3 p.m.) results in greater improvements in insulin sensitivity, oxidative stress, and beta-cell responsiveness compared to a control diet with a 12 h feeding window (07:00 to 19:00), suggesting that the eating period must be in alignment with circadian rhythms to positively affect metabolic outcomes [41].

However, this observation was not replicated in our study, indicating that diet composition, rather than eating time, leads to an improvement in adipokines. A key finding of the present study is that the adoption of AOF resulted in a significant increase in the daily intake of MUFA, which remained unchanged during TRE. The MUFA component of the MedDiet has been shown to mediate its effects on cardiovascular risk factors by reducing the subclasses and fractions of low-density lipoprotein cholesterol, leading to an antiatherogenic lipid profile [42]. Previous work has demonstrated that a MUFA-rich diet can prevent central fat distribution and down-regulate postprandial adiponectin expression in insulin-resistant subjects [43]. Nasir et al. have shown that MUFA consumption is significantly and positively associated with plasma omentin concentrations among women with obesity [44]. Therefore, it could be hypothesized that increased MUFA intake is a driver of the positive effects of AOF on vaspin and omentin circulating levels.

Another interesting finding of this study is that omentin concentrations in the AOF group increased significantly at 12 weeks compared to both baseline and 7 weeks, 5 weeks after participants had in theory stopped fasting and returned to their normal eating plans. This observation could be attributed to the fact that Orthodox fasters preferred not to completely abandon the diet, but to incorporate some of its elements into their regular nutritional plans, motivated by their religious beliefs. Another possible reason for continuing the Athonian diet even in non-fasting periods is that religious fasting has been associated with improvements in emotional stress and mental health, mediated by changes in the psychoneuroendocrine environment [45]. With respect to OF in particular, it has been shown to have an independent positive impact on cognition and mood, a finding that could be related to the large amount of antioxidants consumed during fasting periods [46].

The clinical implications of the effects of AOF on vaspin and omentin concentrations are unclear at the moment. Hao et al. have shown that elevated circulating vaspin levels are not only positively correlated with BMI in people with T2D, but also with the prevalence of coronary artery disease (CAD) in this population [47]. T2D is characterized by up-regulation of the vaspin gene; in contrast, vaspin mRNA expression is not detectable in lean subjects with a normal glucose tolerance [48]. Given the cross-sectional character of the most relevant studies, it is difficult to conclude if the increase in vaspin levels simply represents a counter-regulatory mechanism to insulin resistance or if it is causally associated with metabolic disturbances, and thus if by decreasing circulating vaspin an improvement in cardiometabolic outcomes is feasible. Decreased omentin levels are an independent predictor of CAD and are associated with the severity of the disease [49]. In animal models of atherosclerosis, omentin has been shown to decrease macrophage infiltration, pro-inflammatory gene expression, and suppress foam cell formation [50,51]. Therefore, the potential of AOF to increase circulating omentin should, in theory, contribute to the cardioprotective effects of this nutritional advocacy. However, this remains to be tested and confirmed in adequately powered studies in humans.

The strengths of the present study lie in its prospective character and the assessment of adipokine levels and anthropometry at multiple time points. On the other hand, the results should be interpreted in light of some limitations. A larger sample size could have enhanced the potential of the study to detect associations between the parameters examined, including the correlation between MUFA intake and changes in vaspin and omentin levels. Although we did not use an objective method to quantify the physical activity of the participants, the fact that both groups received specific guidance on the desired kind and type of exercise they should maintain during the study should alleviate the effect of physical activity on adipokine levels. Ideally, the participants should have been randomly assigned to the two diets. However, this would not be feasible without violating the religious beliefs of the Orthodox fasters. Finally, body composition was assessed with BIA and not dual-energy X-ray absorptiometry, which is the gold standard method.

## 5. Conclusions

In conclusion, our results suggest that AOF, a nutritional pattern that combines the MedDiet principles with calorie restriction and TRE, improves vaspin and omentin concentrations among overweight individuals. Future studies should shed more light on relative mechanisms and explore the implications of these findings with respect to cardiometabolic outcomes, such as cardiovascular disease and diabetes.

## Figures and Tables

**Figure 1 nutrients-15-05058-f001:**
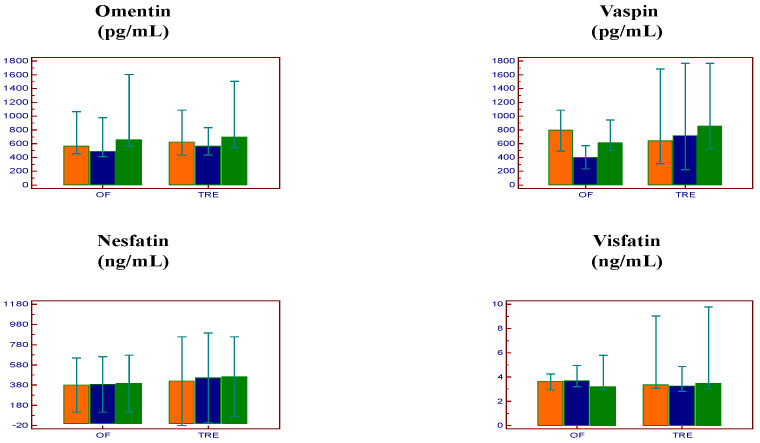
Changes in adipokine levels in the two groups during the study. Orange: Baseline; Blue: Week 7; Green: Week 12; Data are presented as medians (95% confidence interval for median) (omentin, vaspin, and visfatin) and as means (95% confidence interval for mean) (nesfatin) OF: Orthodox fasting group; TRE: time-restricted eating group.

**Table 1 nutrients-15-05058-t001:** Changes in anthropometric, biochemical and dietary parameters in the two groups during the study.

Parameter	Baseline	Week 7	Week 12	*p* (Baseline-Week 7)	*p* (Week 7-Week 12)	*p* (Baseline-Week 12)
Orthodox Fasters/Time-Restricted Eating						
**Age (years)**	50.3 */47.7	-	-	-	-	-
**Male gender (%)**	24.0/33.3	-	-	-	-	-
**Body Mass Index (kg/m^2^)**	29.3/29.1	28.5/28.3	28.3/28.0	**<0.001 †/0.02**	0.97/0.19	**0.004/<0.001**
**Body fat (%)**	36.3/32.8	35.5/31.8	32.9/30.5	1.00/0.29	**0.001/0.01**	**0.01/0.02**
**Lean body mass (kg)**	47.8/53.2	47.3/53.1	48.8/53.5	1.00/1.00	**0.001**/1.00	0.74/1.00
**Waist circumference (cm)**	93.4/94.9	92.6/91.5	92.0/90.5	0.33/**0.01**	0.18/0.60	0.10/**0.007**
**Omentin (pg/mL)**	568.5/622.1	489.4/567.5	659.0/698.2	0.28/0.77	**0.001**/0.41	**0.001**/0.94
**Vaspin (pg/mL)**	795.8/641.6	402.7/715.4	612.5/855.3	**0.002**/1.00	**0.004**/0.60	1.00/0.90
**Nesfatin (ng/mL)**	380.9/419.5	387.9/452.1	395.1/464.0	0.10/0.95	1.00/1.00	1.00/1.00
**Visfatin (ng/mL)**	3.6/3.4	3.7/3.2	3.2/3.5	1.00/0.19	0.87/0.94	0.39/1.00
**Daily fat intake (g)**	86.9/84.6	70.2/57.0	87.5/78.2	**<0.001/<0.001**	**0.01/0.01**	1.00/0.11
**Daily saturated fat intake (g)**	24.5/25.6	9.5/13.4	27.0/21.7	**<0.001/<0.001**	**<0.001/0.004**	1.00/0.43
**Daily monounsaturated fat intake (g)**	32.9/35.4	45.6/32.5	32.0/26.0	**<0.001**/1.00	**<0.001**/0.18	1.00/0.23
**Total dietary fiber intake (g)**	30.7/34.0	30.6/28.3	22.7/24.4	1.00/1.00	0.22/0.42	0.40/0.12

* Mean values are presented. † Bold *p*-values represent statistical significance.

## Data Availability

The data presented in the study are available on reasonable request from the corresponding author.

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
