# Peer review of "A Mediterranean Eating Pattern Combining Energy and Time-Restricted Eating Improves Vaspin and Omentin Concentrations Compared to Intermittent Fasting in Overweight Individuals"

_nutrients, 2023, doi:10.3390/nu15245058_

Round 1
Reviewer 1 Report
Comments and Suggestions for Authors
The article investigates the impact of Athonian Orthodox Fasting (AOF), characterized by a combination of energy and time-restricted eating (TRE) based on the Mediterranean diet, compared to another TRE model on adipokine levels such as visfatin, omentin, nesfatin, and resistin. The study involved 25 individuals practicing AOF and 12 participants following the 16:8 TRE model, serving as the control group.
Findings revealed significant decreases in visfatin levels and increases in omentin levels in the AOF group, while the TRE group did not show significant changes in the analyzed adipokines. Conclusions drawn from the study suggest that AOF may have a positive impact on cardiometabolic health by significantly reducing visfatin levels and increasing omentin levels. Importantly, visfatin and omentin are known to be associated with obesity and type 2 diabetes, making these results particularly relevant. The authors appropriately acknowledge the need for further research on the implications of these findings for cardiometabolic health.
However, there are aspects that could be addressed for a more comprehensive understanding. The sample size is relatively small, potentially limiting the generalizability of the results. Additionally, a more detailed discussion of potential mechanisms underlying the observed changes in adipokine levels could enhance the scientific contribution of the study.
In summary, the article presents valuable insights into the potential benefits of AOF compared to the traditional TRE model in terms of adipokine levels associated with obesity and type 2 diabetes. The results are noteworthy and provide a basis for further research on larger and more diverse populations to confirm the observed effects and elucidate the underlying mechanisms.
Introduction:
The introduction to the article provides a comprehensive background review and rationale for the study. However, there are several points that could be addressed to increase clarity and precision:
The introduction provides detailed information on Orthodox Christian fasting (OF) and Athonian Orthodox fasting (AOF) without clearly establishing the specific context or motivation for the current study. It lacks a concise statement of the research question or hypothesis to which the study is intended to respond.
The statement, "Vaspin, omentin, nesfatin, and visfatin are recently discovered adipokines whose levels have been shown to correlate with the risk of developing obesity and type 2 diabetes (T2D)" should include a timeframe for when these adipokines were discovered to provide context for the term "recently." Additionally, specific references confirming these correlations would strengthen this statement.
The transition between discussing AOF and introducing visfatin, omentin, nesfatin, and resistin could be smoother. Consider adding a transitional sentence to clearly connect these two sections and emphasize why these adipokines are significant in the context of fasting.
The transition from discussing specific adipokines to time-restricted eating (TRE) is somewhat abrupt. Providing a smoother transition or a brief explanation of how TRE relates to the study's aim would increase coherence.
The sentence starting with "Together, the data presented above..." could be clearer in directly stating the study's objective and hypothesis.
The expression "...overweight but otherwise metabolically healthy individuals" lacks a clear definition or operationalization. A brief explanation of how "metabolically healthy" is defined in the study context would be beneficial.
Materials and Methods:
In section 2.1, inclusion criteria are stated as "Healthy adults with a body mass index (BMI) between 25 and 29.9 kg/m2..." but it would be helpful to specify the age range, body weight, or height range if applicable.
In section 2.1, it is mentioned that exclusion criteria include "prediabetes/diabetes," but it would be more precise to specify whether individuals with diagnosed diabetes or prediabetes were excluded and the criteria used for exclusion.
In section 2.3, anthropometric measurements are described, but it is not mentioned whether these measurements were taken by a single trained observer or multiple individuals. Measurement consistency is crucial for accuracy, and this information would be relevant for the study's credibility.
In section 2.5, it is stated that "(variables without a normal distribution underwent the logarithmic transformation prior to the analysis)" A brief explanation of the rationale for this transformation and its potential impact on result interpretation would be helpful.
Section 2.6, containing ethical considerations, could be expanded to include information on participant confidentiality and data processing to provide a comprehensive overview of ethical safeguards.
Results:
In section 3.1, when reporting changes in anthropometric parameters, it would be beneficial to include effect sizes or confidence intervals to complement p-values, providing a more comprehensive understanding of the results.
In Table 1, units for "Visfatin" are listed as "mg/ml," but in the text, they are referred to as "ng/ml." Verification and correction of units for consistency are necessary.
In Table 2, units for "Visfatin" are listed as "ng/ml," consistent with the text.
Consider adding error bars to the figures (Fig. 1 and Fig. 2) depicting adipokine levels to visually convey variability and ensure a clearer representation of the data.
Discussion:
When discussing observed changes in visfatin and omentin, it would be valuable to relate these changes to potential clinical implications or significance for metabolic health. How do changes in these adipokines relate to broader health outcomes, and what might be their relevance for individuals with obesity or metabolic disorders?
Consider adding a brief explanation or definition of terms such as "chrononutrition" and "TRE" (time-restricted eating) in the discussion to ensure that readers without specialized knowledge can more easily follow the conversation.
Author Response
Reviewer 1
The article investigates the impact of Athonian Orthodox Fasting (AOF), characterized by a combination of energy and time-restricted eating (TRE) based on the Mediterranean diet, compared to another TRE model on adipokine levels such as visfatin, omentin, nesfatin, and resistin. The study involved 25 individuals practicing AOF and 12 participants following the 16:8 TRE model, serving as the control group.
Findings revealed significant decreases in visfatin levels and increases in omentin levels in the AOF group, while the TRE group did not show significant changes in the analyzed adipokines. Conclusions drawn from the study suggest that AOF may have a positive impact on cardiometabolic health by significantly reducing visfatin levels and increasing omentin levels. Importantly, visfatin and omentin are known to be associated with obesity and type 2 diabetes, making these results particularly relevant. The authors appropriately acknowledge the need for further research on the implications of these findings for cardiometabolic health.
However, there are aspects that could be addressed for a more comprehensive understanding. The sample size is relatively small, potentially limiting the generalizability of the results. Additionally, a more detailed discussion of potential mechanisms underlying the observed changes in adipokine levels could enhance the scientific contribution of the study.
In summary, the article presents valuable insights into the potential benefits of AOF compared to the traditional TRE model in terms of adipokine levels associated with obesity and type 2 diabetes. The results are noteworthy and provide a basis for further research on larger and more diverse populations to confirm the observed effects and elucidate the underlying mechanisms.
Answer: The authors thank the reviewer for the time spent reviewing the manuscript and the comprehensive comments made. We believe that the proposed changes have substantially improved the quality of the manuscript.
Introduction:
The introduction to the article provides a comprehensive background review and rationale for the study. However, there are several points that could be addressed to increase clarity and precision:
The introduction provides detailed information on Orthodox Christian fasting (OF) and Athonian Orthodox fasting (AOF) without clearly establishing the specific context or motivation for the current study. It lacks a concise statement of the research question or hypothesis to which the study is intended to respond.
Answer: We have updated the relevant part of the introduction to more clearly describe the research hypothesis as well as the rationale of the present study.
In particular: “The research hypothesis to be tested in this study is that the composition of the MedDiet is the main driver of its metabolic benefits. A dietary model that has consistently proven positive metabolic effects, such as TRE, would be the ideal control diet to test our hypothesis. However, the comparison between the two nutritional patterns would not be reasonable if MedDiet did not include features of intermittent fasting. For this reason, we chose to compare AOF with TRE, since both diets include eating windows, although with slightly different time intervals (12 versus 8 hours). Therefore, we prospectively assessed the impact of AOF on vaspin, omentin, nesfatin and visfatin levels, compared to a TRE model, in overweight but otherwise metabolically healthy individuals (i.e. without a diagnosis of a chronic metabolic disease requiring medical surveillance).” (L 96-105)
The statement, "Vaspin, omentin, nesfatin, and visfatin are recently discovered adipokines whose levels have been shown to correlate with the risk of developing obesity and type 2 diabetes (T2D)" should include a timeframe for when these adipokines were discovered to provide context for the term "recently." Additionally, specific references confirming these correlations would strengthen this statement.
Answer: Thank you for this valuable comment. We have rephrased this part of the text to include a more specific timeframe for when the adipokines were discovered. Furthermore, we have provided specific references (No 13 & 14) relevant to the correlations between adipokine levels and cardiometabolic disease risk.
In particular: “Vaspin, omentin, nesfatin, and visfatin are adipokines discovered nearly 20 years ago whose levels have been shown to correlate with the risk of developing obesity and type 2 diabetes (T2D) [13,14]” (L 64-67)
The transition between discussing AOF and introducing visfatin, omentin, nesfatin, and resistin could be smoother. Consider adding a transitional sentence to clearly connect these two sections and emphasize why these adipokines are significant in the context of fasting.
Answer: Following the reviewer’s comment, we have added a sentence to better connect these specific sections of the introduction. In particular:
“Following dietary interventions that induce weight loss, such as fasting, circulating adipokine levels can change as a result of respective changes in adipose tissue” (L 63-64)
The transition from discussing specific adipokines to time-restricted eating (TRE) is somewhat abrupt. Providing a smoother transition or a brief explanation of how TRE relates to the study's aim would increase coherence.
Answer: Following the reviewer’s comment, we have added a linking sentence to improve coherence.
In particular: “Considering the close association between adipokine levels and metabolic health, the effects of various dietary models – especially those that are gaining increasing popularity - on these molecules deserve to be investigated” (L 86-88)
The sentence starting with "Together, the data presented above..." could be clearer in directly stating the study's objective and hypothesis.
Answer: The text has been amended as suggested by the reviewer. (L 96-105)
The expression "...overweight but otherwise metabolically healthy individuals" lacks a clear definition or operationalization. A brief explanation of how "metabolically healthy" is defined in the study context would be beneficial.
Answer: In the revised manuscript, we provide a brief explanation of how “metabolically healthy” is defined in the study context.
In particular: “Therefore, we prospectively assessed the impact of AOF on vaspin, omentin, nesfatin, and visfatin levels, compared to a TRE model, in overweight but otherwise metabolically healthy individuals (i.e. without a diagnosis of a chronic metabolic disease requiring medical surveillance).” (L 102-105)
Materials and Methods:
In section 2.1, inclusion criteria are stated as "Healthy adults with a body mass index (BMI) between 25 and 29.9 kg/m2..." but it would be helpful to specify the age range, body weight, or height range if applicable.
Answer: In the revised manuscript, we have provided the age range of individuals who were eligible for enrolment (18-65 years) (L 108). Given that both weight and height are components of the BMI estimation formula, there were no specific ranges for the values of these parameters, but only for the BMI values.
In section 2.1, it is mentioned that exclusion criteria include "prediabetes/diabetes," but it would be more precise to specify whether individuals with diagnosed diabetes or prediabetes were excluded and the criteria used for exclusion.
Answer: In section 2.1 we have provided the specific criteria used for excluding participants with prediabetes or diabetes.
In particular: “Exclusion criteria were: i) presence of chronic kidney disease, severe liver disease, diagnosis of prediabetes (fasting glucose 100-125 mg/dl or glycated hemoglobin 5.7-6.4% or blood glucose 140-199 mg/dl at 2h post 75 g glucose load) or diabetes mellitus (fasting glucose ≥126 mg/dl or glycated hemoglobin ≥6.5% or blood glucose ≥200 mg/dl at 2h post 75 g glucose load)…..” (L 112-116)
In section 2.3, anthropometric measurements are described, but it is not mentioned whether these measurements were taken by a single trained observer or multiple individuals. Measurement consistency is crucial for accuracy, and this information would be relevant for the study's credibility.
Answer: Thank you for this valuable comment. Indeed, all measurements were independently confirmed by a second investigator and this has been clarified in the revised manuscript (L 160-161)
In section 2.5, it is stated that "(variables without a normal distribution underwent the logarithmic transformation prior to the analysis)" A brief explanation of the rationale for this transformation and its potential impact on result interpretation would be helpful.
Answer: We have clarified this issue in the relevant section of the manuscript. In particular: “Since ANOVA represents the parametric statistical test, inclusion of variables that are not normally distributed may produce misleading results. Therefore, variables without a normal distribution underwent the logarithmic transformation prior to the analysis, with the assumption that transforming the data will make them fit the assumptions better.” (L 184-188)
Section 2.6, containing ethical considerations, could be expanded to include information on participant confidentiality and data processing to provide a comprehensive overview of ethical safeguards.
Answer: Thank you for this valuable comment. Section 2.6 has been updated to reflect that all patient data was treated with confidentiality.
In particular: “The personal information of the study participants was processed anonymously to ensure confidentiality” (L 197-198)
Results:
In section 3.1, when reporting changes in anthropometric parameters, it would be beneficial to include effect sizes or confidence intervals to complement p-values, providing a more comprehensive understanding of the results.
Answer: Confidence intervals have been included when reporting the findings in Section 3.1 (L 207-216)
In Table 1, units for "Visfatin" are listed as "mg/ml," but in the text, they are referred to as "ng/ml." Verification and correction of units for consistency are necessary.
Answer: Thank you for pointing this out. The correct units are ng/ml and the typographic error in Table 1 hasbeen corrected.
In Table 2, units for "Visfatin" are listed as "ng/ml," consistent with the text.
Consider adding error bars to the figures (Fig. 1 and Fig. 2) depicting adipokine levels to visually convey variability and ensure a clearer representation of the data.
Answer: Error bars have been added to the figure.
Discussion:
When discussing observed changes in visfatin and omentin, it would be valuable to relate these changes to potential clinical implications or significance for metabolic health. How do changes in these adipokines relate to broader health outcomes, and what might be their relevance for individuals with obesity or metabolic disorders?
Answer: Following the reviewer’s suggestion, we have elaborated in the discussion of the revised manuscript on the potential clinical implications of our findings, focusing specifically on cardiovascular outcomes, which are the main driver of morbidity and mortality in people with diabetes and obesity.
In particular: “The clinical implications of the effects of AOF on vaspin and omentin concentrations are unclear at the moment. Hao et al. have shown that elevated circulating vaspin levels are not only positively correlated with BMI in people with T2D, but also with the prevalence of coronary artery disease (CAD) in this population [47]. T2D is characterized by up-regulation of the vaspin gene; in contrast, vaspin mRNA expression is not detectable in lean subjects with normal glucose tolerance [48]. Given the cross-sectional character of most relevant studies, it is difficult to conclude if the increase in vaspin levels simply represents a counterregulatory mechanism to insulin resistance or if it is causally associated with metabolic disturbances, and thus if by decreasing circulating vaspin an improvement in cardiometabolic outcomes is feasible. Decreased omentin levels are an independent predictor of CAD and are associated with the severity of the disease [49]. In animal models of atherosclerosis, omentin has been shown to decrease macrophage infiltration, pro-inflammatory gene expression, and suppress foam cell formation [50,51]. Therefore, the potential of AOF to increase circulating omentin should in theory contribute to the cardioprotective effects of this nutritional advocacy. However, this remains to be tested and confirmed in adequately powered studies in humans.” (L 321-336)
Consider adding a brief explanation or definition of terms such as "chrononutrition" and "TRE" (time-restricted eating) in the discussion to ensure that readers without specialized knowledge can more easily follow the conversation.
Answer: Thank you for this suggestion. In the discussion of the revised manuscript, we include the definition of the term “chrononutrition”, also providing a relevant reference (No 38).
In particular: “The term “chrononutrition” refers to a relatively new field of health sciences that aims to understand how the time of eating affects human health [38].” (L 283-285)
Regarding TRE, following the comment of the other reviewer, we have included its definition in the introduction of the revised manuscript, also providing a relevant reference (No 27).
In particular: “Time-restricted eating (TRE) is based on the important role of circadian rhythm in physiology and disease, emerging as an alternative to calorie restriction to improve body weight, glucose profile, and cardiovascular risk factors [26]. In TRE, subjects abstain from food for 16 or 20 hours during the day, while subtypes include early TRE (eating early during the day) and late TRE (eating late during the day) [27].” (L 88-93)
Reviewer 2 Report
Comments and Suggestions for Authors
In this manuscript, Karras et al. investigated the effects of the Mediterranean eating pattern combining energy and time-restricted eating compared to intermittent fasting in overweight individuals. The paper is interesting and correctly written, but there are some important issues that should be addressed.
I address some of these below:
Lines 23, 24, and 45 – in the literature, the Mediterranean diet includes poultry and eggs and even red meat on occasion. Please explain why you are referring to the Mediterranean diet when the participants in the AOF group do not practice this type of diet?
Lines 24-24 – the differences in the duration of fasting may account for the differences in the results. Please explain why you chose the comparison between 12 and 8 hours of fasting?
Line 55 – what do the authors mean by optimal anthropometric values? Low fat? A high percentage of free fat mass? BMI?
The introduction should include a brief characteristic of the TRE diet. I suggest a TRE description from the paper: https://doi.org/10.1038/s41598-022-13904-9
Line 93 – “Inclusion of overweight individuals was preferred to improve adherence to diet” – do the authors really mean that overweight individuals are more likely to adhere to the diet than normal-weight individuals? My observation as a dietitian leads to a completely different observation. In my opinion, the motivation to reduce body weight has a more complicated basis than excessive body weight.
Line 177 – What is the reason for the large disparity between the numbers of participants in the groups? This could influence the results.
Line 196 – The color of the text is different
Tables 1 and 2 – the presentation of the results should be improved. It is difficult to understand and compare the results of the two groups when the results are presented in two tables. I suggest to redesign 2 tables into one. It is similar to the presentation of the changes in adipokines. The results of the AOF and TRE groups should be presented in one figure.
Author Response
Reviewer 2
In this manuscript, Karras et al. investigated the effects of the Mediterranean eating pattern combining energy and time-restricted eating compared to intermittent fasting in overweight individuals. The paper is interesting and correctly written, but there are some important issues that should be addressed.
Answer: The authors thank the reviewer for the time spent reviewing the manuscript and the comprehensive comments made. We believe that the proposed changes have substantially improved the quality of the manuscript.
I address some of these below:
Lines 23, 24, and 45 – in the literature, the Mediterranean diet includes poultry and eggs and even red meat on occasion. Please explain why you are referring to the Mediterranean diet when the participants in the AOF group do not practice this type of diet?
Answer: We thank the reviewer for this comment. The main reasons why the Athonian fasting shares several common characteristics with the MedDiet, despite the latter including meat, are the following:
- The protein intake in AOF is largely based on the consumption of fish and seafood, similar to MedDiet, which also typically includes 1-5 servings of fish per week (Azzeh et al. Front Nutr. 2022).
- AOF has a macronutrient composition similar to MedDiet, specifically 50-60 % carbohydrates, 15-20% protein, and 30% fat (Thrichopoulou et al. Forum Nutr. 2005)
- AOF is characterized by a high intake of monounsaturated fat (MUFA) and polyunsaturated fatty acids (MUFA), as happens with MedDiet. High intake of MUFA and PUFA has been shown to mediate the cardiovascular effects of this nutritional pattern.
We have clarified this point in the introduction of the revised manuscript, as suggested by the reviewer. Furthermore, we have cited the relevant studies by Azzeh et al. and Trichopoulou et al. (References No 9 and 10, respectively).
In particular: “The protein intake in AOF is largely based on the consumption of fish and seafood, similar toMedDiet, which typically includes 1-5 servings of fish per week [9]. Furthermore, AOF has a macronutrient composition similar to MedDiet, specifically 50-60% carbohydrates, 15-20% protein, and 30% fat [10]. As we have previously demonstrated, AOF is characterized by a high intake of monounsaturated fat (MUFA) and polyunsaturated fatty acids [6-8], as happens with MedDiet.” (L 51-56)
Lines 24-24 – the differences in the duration of fasting may account for the differences in the results. Please explain why you chose the comparison between 12 and 8 hours of fasting?
Answer: As explained in the introduction of the revised manuscript, the research hypothesis to be tested in this study is that the composition of the MedDiet is the main driver of its metabolic benefits. A dietary model that has proven metabolic benefits, such as TRE, would be the ideal control diet to test our hypothesis. However, the comparison between the two diets would not be reasonable if MedDiet did not include features of intermittent fasting. For this reason, we chose to compare AOF with TRE, since both diets include eating windows, although with slightly different time intervals (12 versus 8 hours).
This important point has been clarified in the introduction of the revised manuscript.
In particular: “The research hypothesis to be tested in this study is that the composition of the MedDiet is the main driver of its metabolic benefits. A dietary model that has consistently proven metabolic benefits,such asTRE, would be the ideal control diet to test our hypothesis. However, the comparison between the two nutritional patterns would not be reasonable if MedDiet did not include features of intermittent fasting. For this reason, we chose to compare AOF with TRE, since both diets include eating windows, although with slightly different time intervals (12 versus 8 hours).” (L 96-102)
Line 55 – what do the authors mean by optimal anthropometric values? Low fat? A high percentage of free fat mass? BMI?
Answer: We have clarified this point in the revised manuscript, as suggested by the reviewer.
In particular: “This is reflected in the fact that compared to lay Orthodox fasters, monks have been shown to consume a lower amount of energy daily [11], resulting in an optimal glucose, lipid and anthropometric [i.e. body mass index (BMI) and body fat (BF) within the normal range] profile, and low insulin resistance [12].” (L 59-62)
The introduction should include a brief characteristic of the TRE diet. I suggest a TRE description from the paper: https://doi.org/10.1038/s41598-022-13904-9
Answer: In the introduction of the revised manuscript, we provide a definition of TRE according to the study by Domaszewski et al. which has been cited (Reference No 27).
In particular: “In TRE, subjects abstain from food for 16 or 20 hours during the day, while subtypes include early TRE (eating early during the day) and late TRE (eating late during the day) [27].” (L 91-93)
Line 93 – “Inclusion of overweight individuals was preferred to improve adherence to diet” – do the authors really mean that overweight individuals are more likely to adhere to the diet than normal-weight individuals? My observation as a dietitian leads to a completely different observation. In my opinion, the motivation to reduce body weight has a more complicated basis than excessive body weight.
Answer: We have rephrased the relevant part of the text to reflect that overweight participants would be more likely to be motivated to be included in the study (and not more adherent, as correctly noted by the reviewer). In particular, according to the systematic review and meta-analysis by Silva et al. published in 2018, better health is one of the main motivations for people to improve their weight. Therefore, we hypothesized that overweight individuals would be more motivated than normal weight peers to improve their health through diet. We have clarified this point in the revised manuscript and also cited the meta-analysis by Silva et al. (Reference No. 29).
In particular: “According to the systematic review and meta-analysis by Silva et al. [29], better health is one of the main motivations for people to improve their weight. Therefore, we hypothesized that overweight individuals would be more motivated than normal weight peers to improve their health through diet [30]” (L 109-112)
Line 177 – What is the reason for the large disparity between the numbers of participants in the groups? This could influence the results.
Answer: Thank you for this comment. TRE is a dietary model that is difficult to adhere to for a long period of time. This is why participants in this group were reasonably fewer than those included in the Athonian fasting group. However, unequal sample sizes in independent groups have no impact on the results and do not influence inference (BMC Res Notes, 2017 Sep 6;10(1):446).
Line 196 – The color of the text is different
Answer: Thank you for pointing this out. The color of the text has been homogenized.
Tables 1 and 2 – the presentation of the results should be improved. It is difficult to understand and compare the results of the two groups when the results are presented in two tables. I suggest to redesign 2 tables into one. It is similar to the presentation of the changes in adipokines. The results of the AOF and TRE groups should be presented in one figure.
Answer: Following the reviewer's suggestion, we have merged the two tables and figures into one table and afigure, respectively, to allow comparisons between the two groups.